# Safety and Immunogenicity of COVID-19 BBIBP-CorV Vaccine in Children 3–12 Years Old

**DOI:** 10.3390/vaccines10040586

**Published:** 2022-04-11

**Authors:** Khaled Greish, Abdulla Alawadhi, Ahmed Jaradat, Amer Almarabheh, Marwa AlMadhi, Jaleela Jawad, Basma Alsaffar, Ejlal Alalawi, Adel Alsayyad, Afaf Merza, Batool Alalawi, Donia Qayed, Ahmed Humaidan, Manaf Al Qahtani

**Affiliations:** 1College of Medicine and Medical Sciences, Arabian Gulf University, Manama 329, Bahrain; ahmedakj@agu.edu.bh (A.J.); amerjka@agu.edu.bh (A.A.); 2Bahrain Defence Force Hospital, West Riffa, Riffa P.O. Box 28743, Bahrain; a.i.a.bh9@gmail.com (A.A.); donia.ali@bdfmedical.org (D.Q.); 3National Taskforce for Combating the Coronavirus (COVID-19), Manama 329, Bahrain; marwa_9811@hotmail.com (M.A.); jjawad@health.gov.bh (J.J.); ahmed.humaidan@pmo.gov.bh (A.H.); 4School of Medical Sciences, University of Manchester, Manchester M13 9NT, UK; 5Supreme Council of Health, Manama 329, Bahrain; 6Ministry of Health, Sanabis 410, Bahrain; bsaffar@health.gov.bh (B.A.); efalawi@health.gov.bh (E.A.); asayyad@health.gov.bh (A.A.); aali9@health.gov.bh (A.M.); balawi1@health.gov.bh (B.A.); 7Royal Collage of Surgeons, Ireland Medical University of Bahrain, Busaiteen 228, Bahrain

**Keywords:** COVID-19, BBIBP-CorV, children 3–12 years old, the anti-spike, anti-nucleocapsid, neutralizing antibody

## Abstract

Background and Objectives: In the current COVID-19 pandemic, children below the age of 12 could manifest COVID-19 symptoms and serve as a reservoir for the virus in the community. The present study was conducted to evaluate the reactogenicity, and immunogenicity of BBIBP-CorV, prior to involving this age group in the vaccination program in the kingdom of Bahrain. Subjects and Methods: The study included 582 children from 3 to 12 years old of Bahraini and non-Bahraini nationality, all of which contributed to the reactogenicity study. Of those, 401 contributed to the immunogenicity study. All children received 2 doses of BBIBP-CorV inactivated virus 3 weeks apart. To assess reactogenicity, children were followed up for 5 weeks to evaluate any vaccine-related adverse events (AE). To assess immunogenicity, blood was collected on day 0 and day 35 to assess antibody titer against S, N, and neutralizing antibody. Results: Of the 582 participants, (45.4%) were female, (54.61%) were male, with 49% in 9–12 age group. Of the 401 children contributing to the immunogenicity study, 274 (68.3%) had no prior exposure to COVID-19. The overall incidence of AE was 27.7%. No significant difference was found among different age groups. The most frequent AE was local (at the injection site) and occurred in 16% of children, followed by fever in 9.3%. No serious adverse events were reported. The Seroconversion rate was 100% among children with no prior exposure to COVID-19. Children with previous COVID-19 exposure had higher averages of anti-S (2379 U/mL compared to 409.1), anti-N (177.6 U/mL compared to 30.9) and neutralizing antibody (93.7 U/mL compared to 77.1) than children with no prior exposure at day 35. Conclusions: Two doses of COVID-19 BBIBP-CorV on the subjects aged between 3 to 12 has good safety and tolerance and can induce an effective immune response and neutralizing antibody titer.

## 1. Introduction

The COVID-19 pandemic continues to cause severe morbidity and mortality globally, resulting in more than 5.6 million deaths as of January 2022. In the Kingdom of Bahrain 330,621 cases were reported, of which 1399 were reported dead due to COVID-19 complications [1]. Vaccines approved by the World Health Organization (WHO) have been widely used as an active control measure in the current pandemic. In this respect, SARS-CoV-2, BBIBP-CorV is a β-propiolactone-inactivated, aluminum hydroxide-adjuvanted COVID-19 vaccine developed by Beijing Institute of Biological Products of Sinopharm CNBG, China. The vaccine is WHO approved and authorized by 45 countries for adults ≥ 18 years [2]. In a multi-center phase 3 randomized clinical trial performed in 2020, the vaccine proved to have an efficacy of 78.1% against COVID-19 infection [3].

The National Health Regulatory Authority (NHRA) of Bahrain, approved the COVID-19 vaccine (SARS-CoV-2, BBIBP-CorV), Beijing strain, for emergency use in adults above 18 years on 15 December 2020, and in August 2021 for the age group 12 to 17 years.

BBIBP-CorV was the first approved COVID-19 vaccine, which was followed shortly by the approval of Pfizer-BioNTech COVID-19 vaccine, Sputnik V, and Oxford–AstraZeneca COVID 19 vaccine.

In Bahrain, children under 14 years represent 20.1% of the population [4]. Available data on COVID-19 infection suggest that affected children usually present with a mild cough and fever symptoms, with the incidence of COVID-19 in children ranging between 1–2%, globally [5,6,7].

However, this age group remains a potential threat to transmit the disease by functioning as a reservoir for the virus in an immunized community, possibly causing breakthrough infection in high-risk groups. The circulation of the virus in children could hypothetically enhance the chance of viral mutations, jeopardizing the value of ongoing vaccination programs. Further, little is known about the chronic complications of COVID-19 infection in children [8].

The current study aimed at exploring the safety and immunogenicity of BBIBP-CorV in children aged 3–12 years in the Kingdom of Bahrain. The study was carried out in anticipation of the primary schools resuming full day classes in Bahrain.

## 2. Methods

### 2.1. Study Design and Participants

The study is an observational cohort study, carried out between August and October 2021, to assess the immunogenetic and reactogenicity effects of participants vaccinated with two doses of BBIBP-CorV. The study was approved by the National Health Regulatory Authority (NHRA) (Approval Code: CRT-COVID2021-154). After obtaining informed consent from the legal guardians, 582 participants were recruited from the ages of 3–12 years of both sex who had no history of positive PCR COVID-19 test or manifestations suggestive of COVID-19 symptoms. Children with a history of bleeding diathesis or conditions associated with prolonged bleeding were excluded. The participants were followed for 35 days after receiving the first dose of BBIBP-CorV for adverse events through regular interviews and phone calls.

### 2.2. Procedures

The potential participants were first screened for eligibility criteria (age 3–12, no history of positive PCR test, no bleeding diathesis or conditions causing prolonged bleeding). Eligible subjects underwent physical examination, including cardiac and respiratory check-ups. Blood samples were collected for the immunogenicity base line evaluation at day 0.

The vaccine was then prepared and administered intramuscularly into the deltoid muscle of the upper arm (4 μg/0.5 mL). All subjects were observed for 20 min in the healthcare facility for any immediate adverse event following immunization. All participants were instructed to report any side effects through a dedicated electronic system. On days 1, 5, and 14, the research team conducted a follow-up call to investigate the presence of adverse events (AE). The second dose was administered on day 21, prior to which all subjects underwent face-to-face interviews for any side effects. On day 22 and day 28, other phone calls were conducted to follow up any AE after the second dose. Face-to-face interviews were conducted again on day 35 during collection of the second blood sample for evaluation of the antibody responses to vaccination. The study was conducted in primary health care centers and the national conventional center.

### 2.3. Immunogenicity

The antigen-specific humoral immune response was analyzed using one commercial immunoassay (S, N) and one pseudo virus neutralization assay (sVNT) before the receipt of the first dose and 14 days after the receipt of the second dose (day 35). S and N antigen-specific humoral immune response were analyzed using The Elecsys Anti-SARS-CoV-2 S assay (Roche Diagnostics GmbH, Mannheim, Germany), which is an electrochemiluminescence immunoassay (ECLIA) that detects IgG antibodies to the SARS-CoV-2 spike protein receptor-binding domain (RBD) on the Cobas e411 module. According to the manufacturer, the measuring range spanned from 0.4 U/mL to 25,000 U/mL (up to 250 U/mL with onboard 1:10 dilution and more than 2500 with onboard 1:100 dilution). Values higher than 0.8 BAU/mL were considered positive. U/mL and BAU (International OMS standard) correlation are U = 0.972 BAU. The pseudovirus sVNT neutralization was analyzed using cPass SARS-CoV-2 Neutralization Antibody Detection Kit (Genscript Biotech Corporation, Nanjing, China); neutralizing antibodies were calculated as 30% inhibitory dose (neutralizing titer 30, NT30).

### 2.4. Outcomes

The current study had 2 main outcomes. The primary outcome was to measure the safety of BBIBP-CorV in children (3–12 years old). The participants were followed up for 35 days after the first dose of vaccination. The adverse events were classified into 4 groups, fever, local reaction (hardness, itch, pain, warmth, redness, swelling), systemic reactions (headache, joint pain, malaise, muscle aches, nausea, vomiting, diarrhea, turbid or red urine), and other adverse events.

The secondary outcome for this study was to evaluate the humoral immunogenicity of the vaccine. Both SARS-CoV-2 Anti S and SARS-CoV-2 anti-N antibodies were measured before vaccination and at 5 weeks after vaccination. The SARS-CoV-2 neutralizing assay was done at the end of the study. The mean titers and seroconversion rates were measured.

### 2.5. Statistical Analysis

The data was analyzed using the Statistical Package for the Social Sciences (SPSS) version 28 (IBM SPSS Statistics, version 28 (IBM Corp., Armonk, NY, USA)). Descriptive statistics were used for qualitative parameters (gender, frequency of physical examination findings, and frequency of adverse events. Chi-square (χ2) test was used to assess the association between categorical variables. The secondary outcome of IgG concentrations of S, N, and neutralizing antibodies in children with or without previous COVID-19 exposure was reported as mean and 95% confidence interval. Two independent samples t-test and analysis of variance (ANOVA) were used to test the significant differences in the means of quantitative variables. For all statistical results, *p* < 0.05 was considered statistically significant.

## 3. Results

The study took place from August to October 2021, and 582 eligible participants were enrolled. Out of this group, 401 participants completed the serological testing on day 0 and day 35. All participants received 2 doses of 0.5 mL of BBIBP-CorV at days 0 and 21. Table 1 summarizes the demographics of participants in the study.

A total of 256 adverse events (AE) were reported in 161 subjects (27.7%). The main AE was related to pain in the injection site (16%), followed by fever (9.3%). No severe adverse events were reported across the study duration. Overall, the most AE were reported in participants aged 9–12 years, however there was no statistical difference regarding incidence of AE by age groups or sex.

Table 2 summarizes the incidence of different AE in the study population.

Overall, 256 adverse events were reported in 161 subjects. The majority of the 256 AE were reported one day following vaccination (82 events in the first day following the first dose and 79 events in the first day following the second dose) (Table 3).

We analyzed the AE in the 401 participants who contributed to the immunogenicity study. Table 4 shows the incidence of AE in children who showed antibody response due to previous COVID-19 exposure, and in those with negative antibody results. No statistically significant difference was found with adverse events in subjects with previous COVID-19 exposure compared to non-exposed subjects, as in Table 4.

### Immunogenicity

Of the 401 individuals who completed the immunogenicity study, 127 were found to have anti-S and anti-N levels indicative of a prior history of COVID-19 even though with no history of symptoms or positive PCR test. None of the cases with prior COVID-19 exposure had any symptoms related to the disease, and the exposure was incidentally discovered during the study. All COVID naïve cases with negative anti-S and anti-N (274) were seroconverted (100%) at day 35.

The mean anti-S increased across the whole age groups with an overall 6-fold increase for all age groups, while the mean anti-N increased by 2.8 fold at D 35 after vaccination.

Table 5 shows the antibody response for the overall tested participants stratified by sex for all age groups.

Table 6a shows the anti-s response according to sex, age group, and previous exposure. Both males and females had comparable responses with no statistically significant differences. Significant difference in anti-S response was found in the age group 6–9 years which had higher antibody response compared to age group 9–12 years, (*p* ≤ 0.05). Further, the antibody response was significantly higher in the group with the previous COVID-19 exposure (*p* ≤ 0.001).

Table 6b shows the anti-N response according to sex, age group, and previous exposure. No difference in anti-N response was found in relation to sex or age group. However, the antibody response was higher in the group with the previous exposure, and the difference was statistically significant.

Similarly, neutralizing antibody response with exposure showed a statistically significant difference favoring a higher response in subjects with earlier exposure (Table 6c).

## 4. Discussion

Coronavirus (SARS-CoV-2), causing severe acute respiratory syndrome, established itself as the worst pandemic in the 21st century [9].

The virus affected the current way of life across the globe. The disease morbidity in terms of its negative impact on quality of life was as deleterious as its mortality toll that exceeded 5 million lives lost over about 2 years.

Widespread vaccination programs utilizing vaccines with proven safety and efficacy is essential to reduce virus transmission and minimize disease-associated morbidity and mortality.

COVID-19 morbidity and mortality were age-dependent and exponentially increased with age, among other risk factors such as the presence of comorbid conditions and obesity [10].

BBIBP-CorV is a product of Sinopharm of the HB02 strain [11] The vaccine is manufactured by isolating and propagating the strain in a Vero cell line followed by inactivation utilizing β-propiolactone. The inactivated viral particles were then adsorbed to aluminum adjuvant and administered as two intramuscular injections every 3 weeks. The kingdom of Bahrain contributed to the multinational study that proved the safety and efficacy of the vaccine in a study involving 40,382 adult participants [3]. The vaccine showed an efficacy of 78.1% and was well-tolerated in the tested population. In May 2021, the WHO Advisory Group of Experts on Immunization (SAGE) recommended using the vaccine as one of the WHO recognized SARS-CoV-2 recommended vaccines [12].

Aggregated reports indicate that a lower infection rate, and decreased occurrence of COVID-19-associated hospitalization is associated with young age groups; however, severe disease still occurs in all age groups [13]. Furthermore, infected children at a young age can serve as a reservoir for the virus in an otherwise vaccinated population. In addition, the circulation of the unmanifested viral infection may play a role in the possible emergence of new variants of the virus. Hence, it is important to consider younger age groups in assessments of COVID-19 spread and consequently incorporating these ages in strategies and policies targeting COVID-19 spread.

In this study, we performed a cohort observational study on 582 volunteers to assess the safety and immunogenicity of BBIBP-CorV in participants aged 3–12 years. All participants received 4 mcg of inactivated particles on day 0 and 21. As shown in Table 2, the most frequent vaccine-related AE were correlated to pain at the injection site followed by fever. All AEs were mild and resolved spontaneously. Most of the participants (72.3%) did not demonstrate any AE. AEs were reported amongst the cohort for both dose administrations, with more AEs reported after the second dose. Most of the reported AEs emerged on the first day following vaccination, which was observed for both doses. No serious adverse events were reported in any subject in this study. The inactivated vaccine BBIBP-CorV resulted in 100% seroconversion rate in all subjects with no prior exposure to SARS-CoV-2 and augmented the anti-S, anti-N and neutralizing antibodies in those with previous COVID-19 exposure (4.4-fold higher response to anti-S and 2.1-fold higher response to anti-N) Table A3b.

All age groups showed a significant antibody response (S, N, and neutralizing antibody), with the age group 6–9 years showing a significantly higher antibody response against S antigen (anti-S). The AEs were comparable in those with previous infection and virus naive children.

Inactivated BBIBP-CorV vaccine utilizes the same technology known for decades in the management of influenza, polio, and measles [11]. Common to all vaccines in this group, the predominant side effects are the results of local reactions at the injection site as pain, tenderness, swelling, and mild fever attributed to the presence of adjuvants in the vaccine. Optimal dosage requires fractionation of the dose to reduce the vaccine’s unwanted reactions and boost the immune response [14].

In this study, the dose of the vaccine used was the same dose that was administered to an adult patient (4 μg/0.5 mL). Different dosages of BBIBP-CorV had been tested earlier with the dosage of 2,4 and 8 μg in children 3–17 years old in China [15]. The 2 μg dose showed a lower antibody level than the 4 and 8 μg. Between 4 and 8 μg, 4 μg was recommended as it was associated with fewer adverse effects and comparable immunogenic response. Our results showed similar findings to the results reported; namely, the seroconversion rate was 100%, and the vaccine was well tolerated at 4 mg dose. There are several limitations of our study including short duration of the follow up. Longer duration of observation would allow for assessment of the protective effect of the vaccine and allow for the evaluation of the duration of the antibody response. The extension of vaccination programs to involve younger age groups is being extensively studied with a wide variety of other vaccines. Previous work has shown acceptable safety and 100% efficacy with Pfizer BNT162b2 in the age group of 12–25 years old [16]. Similarly, Moderna mRNA-1273 was reported to have 100% efficacy after 2 doses in 12–17 years [17]. Given all reported data on vaccine safety and efficacy in children, it seems plausible that targeting this age group in ongoing SARS-CoV-2 vaccination programs could prove highly beneficial with limited or negligible risk when proven and tested vaccines are utilized in this age group. The continual emergence of new virus variants further highlights the significance of achieving this.

Although the current study sheds light on the safety and immunogenicity of the BBIBP-CorV in children, the study had its limitations given the short follow-up period of 35 days. A longer follow-up would be perceived to determine the vaccine’s efficacy and the duration of antibody response. This study was not a randomized trial and hence representations of the general population could have been better accomplished. Further, the study did not assess the cellular response to the vaccine, which may prove beneficial in illuminating the role of BBIBP-CorV in combating the emerging Omicron variant which is causing global concern. A recent preliminary study on the role of BBIBP-CorV against the hypermutated Omicron was conducted by studying the produced neutralizing antibodies after 2 doses and after 3 doses [18]. In 80% of the vaccinated population the antibody titer against Omicron variants was below detection limit after the second dose. However, the entire study population showed naturalizing antibody presence after the booster dose.

Future work may provide further insight into the role of BBIBP-CorV in protecting emerging variants of SARS-CoV2 through the ongoing surveillance program as the landscape of the virus’s variants evolves.

## 5. Conclusions

In conclusion, this study demonstrated the value of BBIBP-CorV in providing a favorable immunogenic response in 100% of tested subjects involving both S, N antigens, and neutralizing antibodies, while the reported adverse effects were all tolerated. The current study can provide insight for large-scale use of BBIBP-CorV in children aged 3–12 years old.

## Figures and Tables

**Table 1 vaccines-10-00586-t001:** The age and sex of participants.

		Reactogenicity		Immunogenicity	
		No.	%	No.	%
**Gender**	Female	264	45.4	188	46.9
	Male	318	54.6	213	53.1
**Age**	3–<6	93	16.0	60	15.0
	6–<9	204	35.1	147	36.6
	9–<12	285	49.0	194	48.4
**Total**		582	100.0	401	100.0

**Table 2 vaccines-10-00586-t002:** Analysis of the incidence of adverse events after the two doses of vaccination.

Items	Age Group	Total No.	Events	No. of Subjects	Incidence %	Chi-Square *p* Value
**Local Adverse Events (injection site)**	3–<6	93	11	10	10.8	0.320
	6–<9	204	39	34	16.7	
	9–<12	285	60	49	17.2	
**Total**		582	110	93	16.0	
**Systemic Adverse Events**	3–<6	93	5	5	5.4	0.892
	6–<9	204	11	11	5.4	
	9–<12	285	20	18	6.3	
**Total**		582	36	34	5.8	
**Fever**	3–<6	93	14	13	14.0	0.194
	6–<9	204	28	20	9.8	
	9–<12	285	30	21	7.4	
**Total**		582	72	54	9.3	
**Other Adverse Events**	3–<6	93	7	7	7.5	0.199
	6–<9	204	19	16	7.8	
	9–<12	285	12	12	4.2	
**Total**		582	38	35	6.0	
**Overall AE**	3–<6	93	37	23	24.7	0.522
	6–<9	204	97	62	30.4	
	9–<12	285	122	76	26.7	
**Total**		582	256	161	27.7	

**Table 3 vaccines-10-00586-t003:** Incidence rate of adverse events at each time point after vaccination.

	Adverse Event Number		Total
	Dose 1	Dose 2	
**Day 0**	2		2
**Day 1**	82		82
**Day 7**	15		15
**Day 14**	8		8
**Day 21**	6	2	8
**Day 22**		79	79
**Day 28**		36	36
**Day 35**		26	26
**Total**	**113**	**143**	**256**

**Table 4 vaccines-10-00586-t004:** Comparison of the adverse events with exposure.

Item			Female		Male		Total		Chi-Square *p* Value
	Age	No of Events	Non Exposed	Exposed	Non Exposed	Exposed	Non Exposed	Exposed	
**Local Adverse Events (injection site)**	3–<6 Year	**10**	5	3	1	1	6	4	0.920
	6–<9 Year	**31**	14	4	6	7	20	11	
	9–<12 Year	**45**	13	4	17	11	30	15	
	**Total**	**86**	**32**	**11**	**24**	**19**	**56**	**30**	
**Systemic Adverse Events**	3–<6 Year	**4**	1	0	3	0	4	0	NA
	6–<9 Year	**6**	4	0	1	1	5	1	
	9–<12 Year	**14**	5	0	5	4	10	4	
	**Total**	**24**	**10**	**0**	**9**	**5**	**19**	**5**	
**Fever**	3–<6 Year	**11**	2	2	4	3	6	5	0.476
	6–<9 Year	**17**	5	1	8	3	13	4	
	9–<12 Year	**16**	5	3	6	2	11	5	
	**Total**	**44**	**12**	**6**	**18**	**8**	**30**	**14**	
**Other Adverse Events**	3–<6 Year	**1**	0	0	1	0	1	0	NA
	6–<9 Year	**9**	2	0	3	4	5	4	
	9–<12 Year	**9**	2	1	4	2	6	3	
	**Total**	**19**	**4**	**1**	**8**	**6**	**12**	**7**	
**Overall AE**	3–<6 Year	**22**	7	4	7	4	14	8	0.965
	6–<9 Year	**59**	23	5	17	14	40	19	
	9–<12 Year	**75**	22	6	29	18	51	24	
	**Total**	**156**	**52**	**15**	**53**	**36**	**105**	**51**	

**Table 5 vaccines-10-00586-t005:** Antibody response in males and females in different age groups.

	Age (Years)							
	3 to <6 Year		6 to <9 Year		9 to <12 Year		Total	
	Female	Male	Female	Male	Female	Male	Female	Male
**Number (%)**	28 (46.7)	22 (53.3)	74 (50.3)	73 (49.7)	86 (44.3)	108 (55.7)	188	213
**Seroconversion rate**	22 (100%)	20 (100%)	49 (100%)	49 (100%)	62 (100%)	72 (100%)	133	141
**Mean value (95% C.I) D0 anti-S**	85.57 (8.98–208.04)	169.73 (71.91–293.60)	197.87 (121.41–277.73)	163.84 (101.59–242.68)	186.20 (94.88–310.35)	172.51 (108.80–246.13)	175.81 (119.65–241.67)	169.12 (126.11–213.98)
**Mean value (95% C.I) D35 anti-S**	866.89 (519.96–1320.12)	1183.81 (862.35–1539.74)	1171.04 (859.02–1498.37)	1245.22 (906.66–1647.42)	898.13 (645.19–1198.56)	919.99 (726.58–1129.98)	1000.9 (819.66–1191.69)	1071.09 (907.74–1252.13)
**Mean value (95% C.I) D0 anti-N**	22.99 (1.03–50.11)	27.74 (10.25–49.11)	29.48 (15.87–45.13)	28.50 (14.68–47.44)	21.83 (10.13–35.01)	30.72 (18.81–43.77)	25.01 (16.70–33.72)	29.51 (21.38–38.76)
**Mean value (95% C.I) D35 anti-N**	64.69 (41.58–90.66)	90.75 (65.53–118.75)	89.79 (73.33–107.03)	80.55 (72.23–86.89)	64.94 (49.06–82.34)	81.74 (72.81–88.69)	74.69 (63.26–84.86)	80.53 (70.19–92.09)
**Mean value (95% C.I) Neutralizing Ab**	82.74 (77.80–87.56)	87.49 (84.11–90.60)	82.37 (78.75–85.79)	85.09 (81.85–88.16)	78.51 (74.22–82.30)	82.13 (79.2–84.82)	80.66 (78.05–82.91)	83.95 (81.99–85.90)

**Table 6 vaccines-10-00586-t006:** (**a**) The comparison of the mean value of anti-S according to age, sex, and exposure. (**b**) The comparison of the mean value of anti-N according to age and sex and exposure. (**c**) The comparison of the mean value of neutralizing antibody with exposure (at day 35).

(**a**)			
**Variables**	**Delta Mean (post–pre)**	**Statistics**	***p* Value**
**Mean value anti- S**			
**Gender** Male (*n* = 213)	901.97 ± 1112.67	T = 0.718	0.473 ^a^
Female (*n* = 188)	825.09 ± 1017.72		
**Age** 3 to < 6 (*n* = 60)	905.46 ± 872.22	F = 3.276	*p* **< 0.05**^b^
6 to < 9 (*n* = 147)	1026.91 ± 1283.83		
9 to < 12 (*n* = 194)	865.93 ± 1068.57		
**Exposure** Negative (*n* = 274)	408.74 ± 413.35	T = 16.018	*p* **< 0.001**^a^
Positive (*n* = 127)	1841.10 ± 1350.13		
(**b**)			
**Variables**	**Delta Mean (post–pre)**	**Statistics**	***p* Value**
**GMT anti-N**			
**Gender** Male (*n* = 213)	51.02 ± 64.63	T = 0.216	0.829 ^a^
Female (*n* = 188)	49.67 ± 59.77		
**Age** 3 to < 6 (*n* = 60)	53.07 ± 57.99	F = 2.425	0.090 ^b^
6 to < 9 (*n* = 147)	58.30 ± 66.07		
9 to < 12 (*n* = 194)	43.57 ± 60.02		
**Exposure** Negative (*n* = 274)	30.83 ± 36.97	T = 10.331	*p* **< 0.001**^a^
Positive (*n* = 127)	92.09 ± 81.84		
(**c**)			
**Variables**	**Mean**	**Statistics**	***p* Value**
**Neutralizing Antibody**			
**Exposure** Negative (*n* = 274)	77.11 ± 15.8	T = 11.507	*p* **< 0.001**^a^
Positive (*n* = 127)	93.70 ± 5.79		

a: independent sample *t*-test b: analysis of variance (ANOVA).

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
