# Peer review of "Safety and Immunogenicity of COVID-19 BBIBP-CorV Vaccine in Children 3–12 Years Old"

_vaccines, 2022, doi:10.3390/vaccines10040586_

Round 1
Reviewer 1 Report
The manuscript provides the results of safety and immunogenicity of COVID-19 BBIBP-CorV vaccine 1 in children 3-12 years in the kingdom of Bahrain. In general, the work was well designed and the authors offer the necessary information for the conclusions obtained. I only have two minor suggestions to make.
The introduction states that the National Health Regulatory Authority (NHRA) of Bahrain, approved the Covid-19 vaccine (SARS-CoV2, BBIBP-CorV), Beijing strain, for emergency use in adults above 18 years on Dec. 15th, 2020, and in August 2021 for the Age group 12 to 17 years. However, the local regulatory agency approved this study in children between 3 and 12 years of age. I suggest that the authors offer more information on this aspect, that is, if this is part of a clinical trial, an intervention study ... and other details that conditioned the conduct of this research that is outside of what is regulated for that vaccine. It would also be interesting to mention whether other vaccines against SARS-CoV2 have been previously administered in that region and, if so, what the results have been.
It is necessary to improve the structure of the paragraphs between lines 194 and 202 in the discussion, avoiding that the paragraphs are a single separate sentence. Please check the text carefully and correct some other parts with the same problem e.g 115-125.
Author Response
We thank the reviewers for their time, and constructive review.
Kindly find the following response to the comments raised by the reviewer:
The manuscript provides the results of safety and immunogenicity of COVID-19 BBIBP-CorV vaccine 1 in children 3-12 years in the kingdom of Bahrain. In general, the work was well designed and the authors offer the necessary information for the conclusions obtained. I only have two minor suggestions to make.
- The introduction states that the National Health Regulatory Authority (NHRA) of Bahrain, approved the Covid-19 vaccine (SARS-CoV2, BBIBP-CorV), Beijing strain, for emergency use in adults above 18 years on Dec. 15th, 2020, and in August 2021 for the Age group 12 to 17 years. However, the local regulatory agency approved this study in children between 3 and 12 years of age. I suggest that the authors offer more information on this aspect, that is, if this is part of a clinical trial, an intervention study ... and other details that conditioned the conduct of this research that is outside of what is regulated for that vaccine.
We thank the reviewer for the comment. This study was requested from the Bahraini government prior to the start of the 2021-2022 school year. The National Taskforce for Combating the Coronavirus was requested to plan and conduct the current study to inform the possible use of the vaccine in primary school children. While Bahrain had used 4 different COVID vaccines in its vaccination campaign, BBIBP-CorV was the vaccine available at this time. As per the reviewer suggestion, we added the following sentence in the introduction: “The study was carried out in anticipation of the primary schools resuming full day classes in Bahrain”.
- It would also be interesting to mention whether other vaccines against SARS-CoV2 have been previously administered in that region and, if so, what the results have been.
We thank the reviewer for the enquiry. In kingdom of Bahrain Sinopharm’s vaccine was the first to be approved. However, this was shortly followed by the approval of vaccines from Pfizer, and Astra Zeneca late in 2020 and Sputnik V vaccines early in 2021.
39% of the population had Sinopharm’s BBIBP-CorV, 16% Pfizer-BioNTech COVID-19 Vaccine, 13% Sputnik V, and 6% had Oxford Astra Zeneca COVID 19 vaccine. The national task force had conducted a post authorization phase 4 study on the safety, immunogenicity, and efficacy of the 4 vaccines soon to be published. Overall, the vaccines efficacy ranged from 82% to 96% for 2 doses of each vaccine in Bahraini population.
The following was added to the text according to the reviewer recommendation:
“BBIBP-CorV was the first approved vaccine, which was followed shortly by the approval of Pfizer-BioNTech COVID-19 Vaccine, Sputnik V, and Oxford Astra Zeneca COVID 19 vaccine”.
- It is necessary to improve the structure of the paragraphs between lines 194 and 202 in the discussion, avoiding that the paragraphs are a single separate sentence. Please check the text carefully and correct some other parts with the same problem e.g 115-125.
As per the reviewer’s recommendations, both paragraphs were rewritten as highlighted in the text.
Reviewer 2 Report
Greish and colleagues assessed the safety and immunogenicity of COVID-19 BBIBP-CorV vaccine in children present in the Kingdom of Bahrain, aged 3-12 years old. The study aimed to evaluate the adverse side effects (reported by the participants and through interview) and immunogenicity (measuring anti-SARS IgG) after two doses of vaccine. This observational study was conducted on 582 children, of those, 401 contributed
to the immunogenicity study. The authors reported that no statistical difference regarding adverse effect in different age groups or sex, most effect appeared on children 9-12. Regarding the vaccine efficacy, the authors showed that the vaccine induce antibodies in all children, with a significant higher level in previously exposed children compared to nonexposed children.
In general the manuscript could be beneficial especially for residents in the kingdom of Bahrain. However, I have some questions
Major points:
1) The authors showed that anti-COVID-19 antibodies was higher in exposed than non exposed children, by measuring the level of antibody at day 35. However, it could be these antibodies induced by previous infection. It is important to measure the level of antibodies in the exposed children at day 0 and normalize the antibody level at day 35 to day 0 to assess the level of antibody induced in the exposed children and compared to non non-exposed group.
2) Did the authors measure the antibody level at day 0 on the non-exposed children?
3) Did the authors measure the antibody level after the first dose and before the start of the second dose? Kinetics of the antibody level is important.
4) Table 3. Adverse Reactions after the 2 doses: It is confused. The no of participants is not matched with the manuscript and it is confusing. Either rephrase or delete
5) The no of participants in tables is non consistent. This makes confusion. For example table 2 and table 4: In table 2: Local adverse effect
(injection site) (n-93), while in table 4 total no expressed these symptoms (n-86). Please apply this notice to all points in the 2 tables.
Minor:
a) please include the name of hospitals where the participants were recruited.
b) page 4 line 142: "Overall, the most AE were reported in participants aged 6-9 years"
However, in table 2: most side effect in participants aged 9-12 years. Please revise carefully.
Author Response
We thank the reviewers for their time, and constructive review.
Kindly find the following response to the comments raised by the reviewer:
Greish and colleagues assessed the safety and immunogenicity of COVID-19 BBIBP-CorV vaccine in children present in the Kingdom of Bahrain, aged 3-12 years old. The study aimed to evaluate the adverse side effects (reported by the participants and through interview) and immunogenicity (measuring anti-SARS IgG) after two doses of vaccine. This observational study was conducted on 582 children, of those, 401 contributed
to the immunogenicity study. The authors reported that no statistical difference regarding adverse effect in different age groups or sex, most effect appeared on children 9-12. Regarding the vaccine efficacy, the authors showed that the vaccine induce antibodies in all children, with a significant higher level in previously exposed children compared to nonexposed children.
In general, the manuscript could be beneficial especially for residents in the kingdom of Bahrain. However, I have some questions
Major points:
1) The authors showed that anti-COVID-19 antibodies was higher in exposed than non exposed children, by measuring the level of antibody at day 35. However, it could be these antibodies induced by previous infection. It is important to measure the level of antibodies in the exposed children at day 0 and normalize the antibody level at day 35 to day 0 to assess the level of antibody induced in the exposed children and compared to non-exposed group.
We thank the reviewer for the comment. As per the reviewer comments, we added 2 tables in supplementary information. Table S3 a shows the antibody response for the children with no prior exposeur and table S3b for children with prior exposeur to COVID-19.
Further we added the following sentence to the discussion:
The inactivated vaccine BBIBP-CorV resulted in 100% seroconversion rate in all subjects with no prior exposure to SARS- CoV2 and augmented the anti-S, anti-N and neutralizing antibodies in those with previous Covid-19 exposure (4.4-fold higher response to anti-S and 2.1-fold higher response to anti-N) Table S3.
2) Did the authors measure the antibody level at day 0 on the non-exposed children?
Yes, as now shown in table S3a
3) Did the authors measure the antibody level after the first dose and before the start of the second dose? Kinetics of the antibody level is important.
We thank the reviewer for this comment. However, we did not measure the response after each dose of vaccination due to the available resource.
4) Table 3. Adverse Reactions after the 2 doses: It is confused. The no of participants is not matched with the manuscript, and it is confusing. Either rephrase or delete
We apologize for the confusion. To clarify this point we added the following:
Overall, 256 adverse events were reported in 161 subjects. The majority of the 256 AE were reported one day following vaccination (82 events in the first day following first dose and 79 events in the first day following the second dose). We also change the title of the table to: Incidence Rate of Adverse Events at Each Time Point after vaccination
5) The no of participants in tables is non consistent. This makes confusion. For example table 2 and table 4: In table 2: Local adverse effect (injection site) (n-93), while in table 4 total no expressed these symptoms (n-86). Please apply this notice to all points in the 2 tables.
Once more, we apologize for the confusion. It is true that the numbers in both tables are different. This is because table 4 represent the adverse events in the 401 participants who accepted to measure their antibody level response before vaccination. We added the following sentence to clarify this point:
“We analyzed the AE in the 401 participants who contributed to the immunogenicity study. Table 4 shows the incidence of AE in children who showed antibody response due to previous COVID-19 exposeur, and in those with negative antibody results”
Minor:
- a) please include the name of hospitals where the participants were recruited.
Added
- b) page 4 line 142: "Overall, the most AE were reported in participants aged 6-9 years"
However, in table 2: most side effect in participants aged 9-12 years. Please revise carefully.
We thank the reviewer for the comment. It is true that the highest number of AE was noticed in the age group 9-12 as correctly mentioned by the reviewer. However, the incidence of AE of the age group 6-9 was higher (30.4 % compared to 26.7%) for the age group 9-12.
We apologize for this oversight. As per the reviewer’s comment, we rephrased this sentence to:
Overall, the most AE were reported in participants aged 9-12 years, however there was no statistical difference regarding the incidence of AE by age groups or sex.
Round 2
Reviewer 2 Report
The authors addressed my questions